# Barriers to Physical Activity Participation Among University Staff: A Cross-Sectional Study

**DOI:** 10.3390/ijerph22071085

**Published:** 2025-07-08

**Authors:** Sami Elmahgoub, Hassan Mohamed, Farah Abu Khadra, Aseel Aburub, Mohamed I. Mabrouk, Adel Eltaguri, Ákos Levente Tóth

**Affiliations:** 1Department of Physiotherapy, Faculty of Applied Health Sciences, Applied Science Private University, Amman 11931, Jordan; a_abualrob@asu.edu.jo (A.A.); m_mabrouk@asu.edu.jo (M.I.M.); 2Department of Sport Training, Faulty of Physical Education and Sport Sciences, University of Tripoli, Tripoli 13932, Libya; has.mohamed@uot.edu.ly; 3Doctoral School of Health Sciences, Faculty of Health Sciences, University of Pécs, 7621 Pécs, Hungary; a014bk@pte.hu; 4Community Medicine Department, Faulty of Medicine, University of Tripoli, Tripoli 13932, Libya; tajoury@doctor.com; 5Institute of Sport Sciences and Physical Education, Faculty of Sciences, University of Pécs, 7624 Pécs, Hungary; tothahu@gmail.com

**Keywords:** physical activity, participation, university staff, barriers

## Abstract

Regular physical activity (PA) is crucial for health, yet many individuals face barriers to engage in an active lifestyle. This study aimed to identify and analyze the barriers preventing university staff from participating in PA. A cross-sectional quantitative approach was utilized, distributing surveys to a diverse sample of 238 university staff aged 19 to 77 years, with an average age of 40. The survey was designed to identify the barriers that individuals face in adhering to physical PA and collected data on various internal and external factors influencing PA participation. Younger participants reported significantly higher scores for lack of energy and motivation compared to older age groups. Additionally, female participants experienced greater internal and external barriers than their male counterparts. Furthermore, university employees experienced significantly higher internal barriers, namely a lack of energy and lack of motivation. The primary barrier to PA participation among university staff was a lack of time. This study highlights the need for supportive environments that address these obstacles to promote PA participation. The findings offer valuable insights for university administrations and policymakers, emphasizing the importance of targeted interventions and supportive policies to enhance the health and activity levels of university staff.

## 1. Introduction

In today’s technology-driven and increasingly sedentary world, physical activity (PA) is essential for maintaining both physical and psychological well-being. More than just structured exercise, PA represents a lifestyle that promotes holistic health by enhancing cardiovascular and muscular strength, improving sleep quality, and reducing the risk of chronic illnesses [1]. By the late 1990s, PA was recognized as a therapeutic tool, and in 2008, it gained prominence under the global initiative “Exercise is Medicine”, emphasizing its role in disease prevention and health promotion across diverse populations [2].

The benefits of PA span multiple organ systems. Thompson et al. has emphasized that no single medication offers as many wide-ranging health benefits as regular PA [3]. These benefits include reduced risks of cardiovascular diseases, type II diabetes, and some cancers, as well as improved mental health through decreased symptoms of depression and anxiety. The World Health Organization (WHO) defines PA as any bodily movement that expends energy, including everyday activities such as walking, cycling, swimming, and participation in team sports [4]. Additionally, PA contributes to improved social interactions, work productivity, and overall life satisfaction [5,6,7].

Despite these established advantages, global PA participation remains suboptimal. The WHO recommends at least 150 min of moderate-intensity or 75 min of vigorous-intensity PA per week to prevent non-communicable diseases and promote overall health [8]. Yet, many individuals fail to meet these guidelines. This widespread inactivity is associated with various health problems, including cardiovascular conditions, obesity, diabetes, and mental health disorders [9,10,11]. These outcomes underscore the urgent need to promote consistent PA for both physical and mental health maintenance [4,12].

Several factors influence PA participation, emphasizing the importance of understanding the barriers that prevent individuals from adopting active lifestyles [13]. These barriers, ranging from personal and behavioral to social and environmental, include competing responsibilities, low motivation, financial constraints, and limited access to facilities [14,15,16]. Additionally, studies have shown that the nature and intensity of these barriers vary across populations. For instance, Safi et al. reported that 80% of workplace participants cited a lack of time and workplace stress as primary obstacles [17], while support from colleagues and access to fitness resources were facilitators. Similarly, academic populations, such as university and high school students, frequently experience decreased motivation and tightly packed schedules that interfere with PA participation [18].

Gender-specific barriers have also been identified. For example, women often face unique challenges related to time management, family responsibilities, body image concerns, and limited facility access. Peng et al. emphasized that sociocultural expectations significantly hinder female engagement in PA, necessitating gender-sensitive fitness programs and supportive environments [19].

University staff, in particular, represent a group that is highly vulnerable to physical inactivity due to demanding schedules, high workloads, and often sedentary job roles [20]. Barriers such as fatigue, lack of motivation, and inadequate access to sports infrastructure further reduce opportunities for PA. Cooper and Barton found that 42% of university employees did not meet recommended PA levels, with women experiencing higher inactivity rates due to time constraints and family obligations [21]. This pattern supports the need for workplace-level interventions to enhance the health and productivity of university employees.

Consistently, studies have shown that time constraints are among the most frequently cited barriers among university faculty and staff. Das et al. reported that university faculty perceive time as the primary obstacle to PA [20], while Borowski et al. found that both time and motivational barriers were most prevalent among employees [22]. Demographic factors, including age and education level, have also been linked to differences in PA engagement. For instance, older and less-educated staff are less likely to meet PA guidelines due to fatigue and lack of access to facilities [23].

Gender disparities in barrier perception are also prominent. Leicht et al. found that female university employees reported more barriers related to time and energy than their male counterparts, suggesting the need for interventions specifically tailored to women’s circumstances [24]. In contrast to barriers, several facilitators have been identified that promote PA participation, including self-efficacy, perceived social support, and access to exercise facilities [24]. Ndupu et al. confirmed that the presence of supportive environments and facility accessibility significantly enhances PA participation among university staff and students [25]. Moreover, cognitive and skill-based limitations can also hinder engagement in PA, reinforcing the need for interventions that consider the specific needs of different occupational groups [25].

For university employees, understanding the barriers to participating in PA is especially important, as they face unique challenges that often lead to inactivity. With more people becoming inactive, addressing barriers to PA among university staff is critical to promoting healthier lifestyles, improving productivity, and reducing healthcare costs. Tailored interventions that address demographic, occupational, and psychosocial differences are necessary to foster inclusive and effective PA strategies across academic institutions.

Despite the global recognition of these issues, this study takes a fresh look at the situation in Libya, where research on PA barriers for university staff is scarce. By identifying these specific challenges, we hope to fill an important gap in the existing literature by investigating the internal and external barriers that hinder PA participation among physically inactive university employees in Tripoli. The findings will provide insights that can help develop targeted interventions and policies to encourage more active lifestyles in academic settings.

## 2. Materials and Methods

### 2.1. Participants

The study involved a total of 238 university staff members (teaching staff, teaching assistants, and employees) out of 335 participants who were identified as physically inactive based on their responses to a screening question regarding their engagement in PA (Do you practice physical activity?). The participants’ ages ranged from 19 to 77 years, with a mean age ± standard deviation of 40 years (mean ± SD: 40 ± 0.978). The sample included 148 females (62%) and 90 males (38%), comprising teaching staff (29%), teaching assistants (2%), and other university employees (69%). Recruitment took place across various sectors of the University of Tripoli, including faculties such as Medicine, Pharmacy, Medical Technology, Economics, Arts, Nursing, Dentistry, Physical Education, Sciences, Languages, Law, Information Technology, and Education, as well as administration buildings like the Clinic Complex Building and University Finance and Administrative Building. Inclusion criteria required participants to be aged 18 years and above and physically inactive, while those who were physically active or under 18 were excluded. This diverse participant pool aimed to accurately represent the barriers faced by university staff in engaging with PA. Ethics approval for this study was obtained from the Bioethics Committee at the Biotechnology Research Center (BEC-BTRC), State of Libya (BEC-BTRC 2-2021).

### 2.2. Data Collection

The investigators distributed self-administered questionnaires to university staff from various sectors of the university, including teaching staff, teaching assistants, and employees. Only those who answered “NO” to the question regarding engagement in physical activity were permitted to continue with the questionnaire. Informed consent was obtained from all participants before participation, ensuring confidentiality and respect for their rights throughout the research process.

Data collection was conducted through face-to-face interviews by trained physical therapy research assistants. They received thorough training on the study procedures and ethical guidelines, which helped them connect well with participants and gather accurate information. The participants either filled out the questionnaires themselves or responded to questions posed by investigators on designated days and times. Participants were provided with the survey, and they were asked to complete it on the spot. All submitted questionnaires were reviewed to ensure that all questions were answered.

### 2.3. Survey

Participants in this study were instructed to complete a questionnaire comprising two distinct sections. The first section collected demographic information, including height, weight, age, occupation, marital status, and engagement in PA. This foundational information is essential for contextualizing the responses provided in the subsequent section. The second section contained 12 questions categorized into two main groups: internal barriers and external barriers to PA participation. The first six questions addressed internal barriers, with each pair of questions focusing on a specific obstacle, while the second set of six questions examined external barriers, again with each pair of questions targeting a particular challenge. Collectively, these questions aim to provide a comprehensive understanding of the internal and external factors that hinder individuals’ participation in PA, thereby facilitating the analysis and identification of various influences on engagement and performance and enriching the overall findings of the study. Responses to the second section were rated on a scale from 0 (does not apply) to 5 (strongly applies). This rating system was utilized to interpret participants’ responses, with scores ranging from 5 (strongly agree) to 1 (strongly disagree). Lower scores indicated negative beliefs, while higher scores reflected positive beliefs.

The questionnaire, developed by Arzu et al. and published online, is freely available for use by other researchers [26]. It was specifically designed to identify barriers that individuals encounter in maintaining regular PA. The internal barriers addressed include energy-related obstacles (Questions 1 and 2), motivational factors (Questions 3 and 4), and self-confidence issues (Questions 5 and 6). In contrast, the external barriers encompass resource availability (Questions 7 and 8), social support (Questions 9 and 10), and time constraints (Questions 11 and 12) [26]. Given its online availability and unrestricted access, this questionnaire has emerged as a valuable instrument for researchers worldwide. It enables exploration of factors influencing exercise behavior across diverse populations and settings. The structured format and clear categorization of questions facilitate administration and analysis, thereby supporting comparative studies and the development of effective strategies to promote PA. The validity of this questionnaire has been confirmed in past studies [26], so we focused on measuring its test–retest reliability in our sample. The test–retest reliability coefficient (Pearson’s correlation coefficient) was equal to 0.89, indicating a strong stability of the scores over time. The average time required for participants to complete the questionnaire ranged from 5 to 15 min, further emphasizing its practicality and accessibility for research purposes.

### 2.4. Statistical Analysis

All data analyses were conducted using SPSS Statistics (Version 23.0; IBM Corp., Armonk, NY, USA). The barrier scores were computed across eleven subscales using a 0–5 Likert Scale, where higher scores indicated stronger agreement with the presence of a barrier. These scores were summarized using means and standard deviations.

To assess group differences in barrier perceptions, we conducted both univariate ANOVA and multivariate analysis of covariance (MANCOVA) tests. The univariate ANOVAs were used to compare barrier scores across demographic groups including age, gender, occupation, BMI, and marital status. Tukey’s HSD post hoc tests were employed where necessary to correct for multiple comparisons. An alpha level of *p* < 0.05 was used to determine statistical significance.

For a more robust multivariate approach, MANCOVA was used to analyze the collective effect of demographic variables (age, sex, occupation, and marital status) on multiple dependent variables (internal and external barriers, energy, and motivation). Assumption checks including Box’s M and Shapiro–Wilk tests were conducted. Pillai’s Trace was used for interpreting multivariate significance due to its robustness to assumption violations.

## 3. Results

### 3.1. General Characteristics of the Study Population

A total of 400 questionnaires were distributed, yielding a substantial response rate of 83.8% with 335 completed surveys returned. Among these respondents, a notable finding emerged: 238 individuals, representing 71% of the total participants, reported a negative response to the question regarding their engagement in PA and consequently participated in this study. The majority of participants are aged 36–50 years (44%) and predominantly female (62%). Most individuals are classified as overweight (36%) or obese (27%). Furthermore, most of the participants were university employees (69%), followed by teaching staff (29%). All participants’ demographic and occupational characteristics are summarized in Table 1.

### 3.2. Barriers to PA Participation for the Whole Study Group

In general, external barriers have higher scores than internal barriers (3.02 ± 0.77 vs. 2.41 ± 0.74; mean ± SD). A lack of time is the most common barrier to participate in PA (3.54 ± 1.19), followed by a lack of resources and lack of motivation (2.98 ± 1.06; 2.55 ± 1.01), respectively. Detailed description about barriers to PA participation among the whole sample are illustrated in Table 2.

### 3.3. Barriers to PA According to Age and Gender

The participants in the young age group (18–35 years) had the highest score in internal barriers (2.66 ± 0.67) compared to the other age groups (2.38 ± 0.74; 2.11 ± 0.71). Notable findings include significant higher score for lack of energy and lack of motives among the young age group than the other age groups. Furthermore, females report significantly higher scores for both internal and external barriers than male participants. More specifically, lack of energy, lack of resources, and lack of time were significantly higher in females. All details regarding barriers to PA participation categorized by age group and gender are displayed in Table 3.

### 3.4. Barriers to PA According to BMI and Occupation

Key findings indicate that there was no significant difference between participants with different BMI categories for all different barriers to PA participation, although external barrier scores in different BMI categories were higher than internal barrier scores. In addition, individuals with obesity report higher lack of energy scores (2.62 ± 1.0) and lack of self-confidence scores (2.4 ± 0.96) compared to normal and overweight groups. Notably, the study revealed significantly higher scores in all internal barriers to PA (except lack of self-confidence) among employees compared to teaching staff and teaching assistant groups (2.52 ± 0.74 vs. 2.19 ± 0.70 and 2.00 ± 0.34), respectively, as shown in Table 4.

### 3.5. Barriers to PA According to Place of Work and Marital Status

The external barriers to PA participation among university staff in different faculties showed higher score than internal barriers, but neither internal nor external barriers scores were found to be significantly different. Single participants had a significant higher score in the internal barriers to PA participation than married participants, namely lack of energy (2.67 ± 1.07 vs. 2.36 ± 0.98), lack of motivation (2.76 ± 1.07 vs. 2.47± 0.96), and all internal barriers (2.62 ± 0.73 vs. 2.33 ± 0.71). Detailed description about barriers to PA participation categorized by place of work and marital status are illustrated in Table 5.

### 3.6. Multivariate Analysis (MANCOVA)

MANCOVA results indicated that age group (Pillai’s Trace = 0.10, F (4, 450) = 5.81, *p* < 0.001) and sex (Pillai’s Trace = 0.07, F (2, 224) = 7.83, *p* < 0.001) had significant effects on combined internal barriers. The interaction between age and sex was not significant (*p* = 0.23), suggesting independent effects of each factor. Table 6 represents results of the MANCOVA. Furthermore, additional MANCOVAs showed that occupation significantly affected the internal barrier variables (Pillai’s Trace = 0.06, F (6, 454) = 2.42, *p* = 0.03), while marital status did not reach significance (Pillai’s Trace = 0.03, F (3, 225) = 2.49, *p* = 0.06), as shown in Table 6.

#### Interpreting Standard Deviation

The standard deviations reflect the variability in participant responses. For example, the SD of 1.19 for time barriers (mean = 3.54) indicates moderate variability, suggesting while time is broadly perceived as a barrier, the strength of this perception varies. Higher SDs for energy and motivation suggest heterogeneous experiences, underlining the importance of tailored interventions.

## 4. Discussion

The research on PA behaviors and barriers in Libya is limited but growing. Studies suggest that PA levels are generally low, with 70% to 75% of the population not meeting recommended PA guidelines. Common barriers include a lack of time, limited facilities, and safety concerns [27,28]. Additionally, cultural norms significantly hinder PA participation among Libyan women due to social expectations and limited opportunities [29]. Previous research highlights a complex interplay of individual, social, and environmental factors affecting PA participation.

In this study, external barriers were more prevalent than internal barriers (3.02 ± 0.77 vs. 2.41 ± 0.74), with lack of time identified as the major obstacle (3.54 ± 1.19). This aligns with previous research that consistently cites lack of time as a significant barrier for busy adults, particularly university employees [22]. Additionally, external barriers such as a lack of resources hinder the participation of university staff in PA, consistent with findings from studies in various context. For instance, a study in South Africa found that limited access to sports facilities was a major obstacle [25]. In contrast, university employees with access to fitness facilities on campus were more likely to participate in regular PA [17]. Studies in the United States and other countries found that proximity to parks and fitness facilities encourages regular PA participation [22]. Furthermore, financial constraints such as the costs of gym memberships and equipment also emerged as significant barriers for university employees. This has been documented in studies from India, Germany, and Iran, emphasizing the need to address financial barriers to promote PA participation [20,29,30]. Moreover, safety concerns and lack of social support were identified as factors that deter individuals from engaging in outdoor PA. Addressing these barriers through strategies like subsidized memberships and community support could foster a more active and healthier workforce [31].

A lack of motivation may diminish both intrinsic desire and external encouragement to engage in regular PA. The absence of a supportive network of physically active individuals can undermine motivation engagement. Studies have shown that having active friends, family members, or colleagues positively influences PA behaviors through encouragement, participation, and shared experiences [32]. This underscores the importance of fostering a culture of PA within university communities.

Strategies such as workplace wellness programs, group exercise classes, and social events can create a supportive environment and overcome the barrier of non-active social network [33]. In addition, fatigue or a lack of energy can also hinder an active lifestyle, while financial constraints related to fitness memberships and equipment can be prohibitive. Limited access to safe facilities, like parks and gyms, also serves as a significant barrier to participation [34].

This study found significant gender differences in the perceived barriers to PA. Female university staff have reported greater internal (2.52 ± 0.68 vs. 2.23 ± 0.78) and external barriers (3.14 ± 0.7 vs. 2.81 ± 0.85) compared to their male counterparts. This aligns with previous research, indicating that women encounter greater challenges with PA participation due to factors like lack of time, family responsibilities, cultural norms, and safety concerns [19]. Similarly, Safi et al. noted that male employees are facing fewer barriers to PA participation compared to female employees [17]. Overall, these findings highlight that woman experience more significant barriers to PA, influenced by various sociocultural, environmental, and personal factors. Understanding these gender-specific barriers is essential for developing targeted interventions to promote PA among female employees. However, the predominance of female participants in our study presents a limitation, as it may affect the generalizability of our results. Future research should aim for a more balanced gender distribution to enhance the applicability of findings across diverse populations.

This study analyzed perceived barriers to PA participation across age groups. Participants aged 18–35 years reported a significantly higher internal barriers (2.66 ± 0.67) compared to the middle-aged (2.38 ± 0.74) and the older-age participants (2.11 ± 0.71). Similarly, the middle-age group showed higher external barriers (3.11 ± 0.77) than the younger (2.98 ± 0.72) and older age groups (2.90 ± 0.83). These findings are consistent with previous research suggesting that age can influence the types of barriers individuals face in engaging in PA. Borowski et al. noted that younger employees were more susceptible to internal barriers, while older employees tend to face more external barriers [22]. This underscores the need for tailored PA interventions: younger employees may benefit from strategies that enhance motivation, while older employees may require support addressing external barriers. However, the majority of participants were less than 50 years, which may not adequately represent the perspectives of the older staff (aged 51 and above). Consequently, the barriers identified may reflect the experiences and challenges specific to younger and middle age groups, potentially overlooking the unique obstacles faced by older employees, who might contend with health-related limitations and different motivational factors. Future research should aim to include a more balanced age distribution to capture a comprehensive view of PA barriers across all age groups within university settings.

Despite there being different barriers scores among university staff with different BMI categories, no significant differences were found. University staff with obesity faced greater deficits in total external and total internal barriers, with key obstacles including a lack of energy, lack of self-confidence, and resources. In contrast, those with normal BMIs cited slightly higher scores for lack of time and motivation. This contrast the findings by Krämer et al., who found that those with higher BMIs were more likely to report a lack of time, lack of motivation, and health concerns as barriers to PA [35]. These mixed findings highlight the need for further research to understand the factors influencing PA participation and to develop targeted interventions for staff across different BMI categories.

University employees encountered significantly higher internal barriers than the other job categories (2.52 ± 0.74 vs. 2.19 ± 0.70 vs. 2.00 ± 0.34). They experience high levels of fatigue after work or other commitments to PA, indicating a lack of energy and motivation to exercise. Specifically, university employees had the highest scores for lack of energy (2.59 ± 1.03) and lack of motivation (2.66 ± 1.02). These findings were consistent with those of Borowski et al., who identified lack of energy as a major barrier to PA, affecting 65% of university employees [22]. Additionally, Cooper et al. also reported that university employees who engaged in less PA also reported higher levels of fatigue and poorer mental health [21]. Furthermore, 55% of participants reported a lack of motivation as a major barrier to PA [36]. This finding emphasizes the importance of addressing individual needs and preferences when promoting PA, as university employees who considered PA enjoyable and beneficial were more likely to engage in PA regularly [20]. Additionally, Ndubu et al. found that 64% of administrative staff and 62% of doctoral students failed to meet recommended PA levels due to lack of knowledge and motivation [25].

The findings of this study indicate that external barriers received the highest barriers scores, ranging from 2.79 ± 0.77 to 3.19 ± 1.01, with lack of time identified as the most significant barrier across faculties. This suggests that the work environment and culture within different faculties influence attitudes toward PA. In contrast, Ndubu et al. reported high levels of physical inactivity among university administrative staff and doctoral students [25]. These differing results underscore the complex factors affecting PA participation in the workplace. While lack of time is a common barrier, the specific organizational culture in academic settings significantly shapes employee attitudes. Further research is needed to explore the interplay between individual, organizational, and cultural factors influencing PA engagement. Targeted interventions addressing the unique barriers in different academic departments may be more effective in promoting a physically active workforce [37].

Single university staff reported significantly higher internal barriers than married university staff (2.62 ± 0.73 vs. 2.33 ± 0.71), particularly for lack of energy and motivation. Conversely, married participants had higher external barriers scores, although the differences were not significant. These findings align with previous research indicating that married individuals face more external barriers, while single individuals encounter more internal challenges [22]. However, some studies suggest that the relationship between marital status and PA barriers is complex and varies by context. For instance, never-married and widowed individuals often engage in less total PA compared to married individuals, with differences in leisure activities; unmarried individuals tend to prefer activities like jogging and weightlifting, while married individuals are more likely to engage in gardening. These disparities may reflect the time demands of marital roles and societal norms regarding appropriate activities for different marital identities [38].

Limited knowledge and awareness, where individuals have a poor understanding of the benefits of PA or misconceptions about exercise, can also impede participation. Physical limitations, such as health conditions, injuries, or disabilities, can pose barriers to engaging in PA. Finally, psychological factors, including negative self-perceptions, body image concerns, or fear of failure, can discourage individuals from participating in PA. Despite these challenges, regular PA is crucial for maintaining physical and mental health, reducing the risk of chronic diseases, and improving overall well-being. Addressing the barriers to PA among university employees is of utmost importance to enhance their health and productivity and create a more supportive and encouraging environment for PA.

In this study, we understand how important it is to think about how our findings can be applied to other groups. Our results offer valuable insights into various factors, such as age, sex, type of occupation, place of work, nutritional status, and marital status. However, it is crucial to remember that our research is specific to its context. The insights we gained may be most relevant to similar university settings. As such, we encourage caution when trying to apply these results to different populations or situations. We hope that future studies will explore these factors further to deepen our understanding across diverse environments.

## 5. Strengths and Limitations

This study tackles an important issue: the participation of university employees in PA and its vital role in promoting health and productivity, especially in today’s sedentary work environments. In addition, this study captures a broad perspective on the barriers to PA by including a diverse range of faculties and job roles. This makes the findings more relevant across the university setting. Furthermore, the questionnaire thoughtfully addresses both internal and external barriers, allowing for a deeper understanding of what hinders PA participation. Importantly, the results align with previous studies, reinforcing their credibility and adding to our knowledge of PA barriers.

However, the study has some limitations. For instance, there is an over-representation of female participants and under-representation of teaching assistant participants, and this may influence the results and interpretations of our findings, making it harder to generalize the findings. Additionally, this study did not take the participants’ health status into consideration, which could affect how widely the findings apply and may overlook significant barriers that individuals with chronic illnesses or disabilities face. Furthermore, since the data rely on self-reported measures, there is a chance that the participants might not accurately reflect their PA levels or the barriers they face. Furthermore, the cross-sectional design of the study also means we cannot establish clear cause-and-effect relationships between the barriers and PA participation. Lastly, as this study has been conducted in Tripoli, Libya, it may not fully capture the experiences of university employees in different cultural or geographic contexts.

## 6. Conclusions

This study identifies significant barriers to PA among university employees, highlighting the influence of factors like age, gender, BMI, marital status, and workplace environment. It finds that external barriers, particularly lack of time and resources, are the most significant, with younger and single participants reporting notable internal barriers. Female participants face higher internal and external challenges than males, including safety concerns and time constraints. Additionally, employees show elevated internal barriers related to energy and motivation, likely exacerbated by workplace demands. The study emphasizes the need for tailored interventions to address these diverse challenges and foster a supportive PA culture, ultimately enhancing employee well-being and productivity.

## Figures and Tables

**Table 1 ijerph-22-01085-t001:** Demographic and occupational characteristics of study population.

Variable	Description	Percentage (*n*= 238)
Age	18–35 years	34%
	36–50 years	44%
	>50 years	23%
Gender	Males	38%
	Females	62%
BMI	Underweight	2%
	Normal	35%
	Overweight	36%
	Obese	27%
Faculty	Health Sciences	35%
	Humanitarian Sciences	34%
	Science and Engineering	8%
	Physical Education	7%
	Others	16%
Occupation	Employees	69%
	Teaching staff	29%
	Teaching assistants	2%
Marital status	Single	68%
	Married	34%

Abbreviations: BMI: body mass index; *n*: sample size.

**Table 2 ijerph-22-01085-t002:** Barriers to PA participation for the whole study group.

Barrier Scores	Mean± SD (*n* = 238)
Lack of energy score	2.45 ± 1.02
Lack of motivation score	2.55 ± 1.01
Lack of self-confidence score	2.25 ± 0.95
**Internal barriers @**	2.41 ± 0.74
Lack of resources score	2.98 ± 1.06
Lack of time score	3.54 ± 1.19
Lack of support score	2.53 ± 0.90
**External barriers @**	3.02 ± 0.77

Abbreviations: **@**: text in bold font represents the total scores of barriers; *n*: sample size; SD: standard deviation.

**Table 3 ijerph-22-01085-t003:** Barriers to PA according to age and gender.

Barrier Scores	Age (*n* = 238)			Gender (*n* = 238)	
	18–35 Yrs Mean ± SD	36–50 Yrs Mean ± SD	>50 Yrs Mean ± SD	Male Mean ± SD	Female Mean ± SD
Lack of energy	**2.79** ± **1.00 ****	2.36 ± 0.99	2.09 ± 0.96	2.07 ± 0.99	**2.67** ± **0.97 ****
Lack of motivation	**2.84** ± **1.01 ****	2.52 ± 0.95	2.17 ± 1.00	2.42 ± 0.90	2.60 ± 0.70
Lack of self-confidence	2.32 ± 0.97	2.27 ± 0.96	2.08 ± 0.93	2.17 ± 0.99	2.29 ± 0.93
**Internal barriers @**	**2.66** ± **0.67 ****	2.38 ± 0.74	2.11 ± 0.71	2.23 ± 0.78	**2.52** ± **0.68 ****
Lack of resources	2.91 ± 1.08	3.08 ± 1.01	2.90 ± 1.15	2.68 ± 1.07	**3.16** ± **1.02 ****
Lack of time	3.56 ± 1.11	3.59 ± 1.14	3.44 ± 1.40	3.32 ± 1.35	**3.68** ± **1.06 ***
Lack of support	2.44 ± 1.01	2.67 ± 0.97	2.37 ± 0.97	2.44 ± 1.01	2.57 ± 0.98
**External barriers @**	2.98 ± 0.72	3.11 ± 0.77	2.90 ± 0.83	2.81 ± 0.85	**3.14** ± **0.70 ****

Abbreviations: **@**: text in bold font represents the total scores of barriers; *n*: sample size; SD: standard deviation; ***** *p* < 0.05; ****** *p* < 0.01.

**Table 4 ijerph-22-01085-t004:** Barriers to PA participation according to BMI and occupation.

Barrier Scores	BMI (*n* = 238)			Occupation (*n* = 238)		
	Normal Mean ± SD	Overweight Mean ± SD	Obese Mean ± SD	Employees Mean ± SD	Teaching Staff Mean ± SD	Teaching Assistant Mean ± SD
Lack of energy	2.40 ± 1.06	2.35 ± 0.98	2.62 ± 1.00	**2.59** ± **1.03 ****	2.13 ± 0.92	2.00 ± 1.06
Lack of motivation	2.54 ± 1.04	2.52 ± 0.90	2.51 ± 1.12	**2.66** ± **1.02 ***	2.33 ± 0.95	1.80 ± 0.76
Lack of self-confidence	2.29 ± 1.07	2.11 ± 0.85	2.40 ± 0.96	2.31 ± 0.97	2.10 ± 0.91	2.20 ± 0.97
**Internal barriers @**	2.41 ± 0.81	2.33 ± 0.69	2.51 ± 0.72	**2.52** ± **0.74 ****	2.19 ± 0.70	2.00 ± 0.34
Lack of resources	2.93 ± 1.09	2.91 ± 1.04	3.15 ± 1.05	3.01 ± 1.11	2.87 ± 0.95	3.70 ± 0.84
Lack of time	3.60 ± 1.17	3.48 ± 1.22	3.54 ± 1.22	3.50 ± 1.17	3.59 ± 1.25	4.40 ± 0.82
Lack of support	2.40 ± 0.90	2.72 ± 0.95	2.46 ± 1.08	2.53 ± 1.00	2.48 ± 0.94	3.00 ± 1.41
**External barriers @**	2.98 ± 0.75	3.04 ± 0.78	3.05 ± 0.82	3.01 ± 0.79	2.98 ± 0.71	3.72 ± 0.75

Abbreviations: **@**: text in bold font represents the total scores of barriers; *n*: sample size, SD: standard deviation; ***** *p* < 0.05; ****** *p* < 0.01.

**Table 5 ijerph-22-01085-t005:** Barriers to PA according to place of work and marital status.

Barrier Scores	Faculty (*n* = 238)					Marital Status (*n* = 238)	
	Hea S. Mean ± SD	Hum S. Mean ± SD	Sc. & En. Mean ± SD	Phy. Edu. Mean ± SD	Others Mean ± SD	Single Mean ± SD	Married Mean ± SD
Lack of energy	2.52 ± 0.97	2.39 ± 1.02	2.33 ± 1.14	2.38 ± 0.92	2.49 ± 1.12	**2.67** ± **1.07 ***	2.36 ± 0.98
Lack of motivation	2.46 ± 0.93	2.48 ± 1.00	2.73 ± 1.21	2.59 ± 1.04	2.76 ± 1.07	**2.76** ± **1.07 ***	2.47 ± 0.96
Lack of self-confidence	2.27 ± 0.93	2.27 ± 0.96	2.33 ± 1.16	2.25 ± 0.95	2.12 ± 0.93	2.41 ± 0.91	2.18 ± 0.97
**Internal barriers @**	2.41 ± 0.70	2.38 ± 0.72	2.46 ± 0.89	2.41 ± 0.76	2.46 ± 0.79	**2.62** ± **0.73 ****	2.33 ± 0.71
Lack of resources	3.16 ± 1.04	2.83 ± 1.08	3.25 ± 1.03	3.06 ± 0.95	2.73 ± 1.11	2.84 ± 1.08	3.06 ± 1.05
Lack of time	3.64 ± 1.14	3.58 ± 1.16	3.60 ± 1.44	3.25 ± 1.30	3.36 ± 1.20	3.49 ± 1.13	3.60 ± 1.19
Lack of support	2.64 ± 0.97	2.55 ± 0.99	2.73 ± 1.11	2.09 ± 0.92	2.31 ± 0.95	2.46 ± 1.09	2.58 ± 0.93
**External barriers @**	3.15 ± 0.71	2.99 ± 0.79	3.19 ± 1.01	2.81 ± 0.69	2.79 ± 0.77	2.93 ± 0.79	3.08 ± 0.74

Abbreviations: **@**: text in bold font represents the total scores of barriers; *n*: sample size; Hea S., Faculty of Health Sciences; Hum S., Faculty of Humanitarian Sciences; Sc. & En., Faculty of Science and Engineering; Phy. Edu., Faculty of Physical Education; SD, standard deviation; * *p* < 0.05; ** *p* < 0.01.

**Table 6 ijerph-22-01085-t006:** Results of the MANCOVA.

Predictor	Pillai’s Trace	F	df (num, den)	*p*-Value
Age and Sex Effects on Internal Barriers				
Age Group	0.10	5.81	4, 450	<0.001
Sex	0.07	7.83	2, 224	<0.001
Age and Sex	0.03	1.26	6, 448	0.23
Occupation and Marital Status Effects on Internal Barriers				
Occupation	0.06	2.42	6, 454	0.03
Marital Status	0.03	2.49	3, 225	0.06
Sex Effects on External Barriers				
Sex	0.06	4.63	3, 227	0.0036

## Data Availability

Data are available upon request to the corresponding author s_elmahgoub@asu.edu.jo.

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
