# Peer review of "Barriers to Physical Activity Participation Among University Staff: A Cross-Sectional Study"

_ijerph, 2025, doi:10.3390/ijerph22071085_

Round 1
Reviewer 1 Report
Comments and Suggestions for Authors
Thank you for the opportunity to review this work. The authors have aimed to examine the internal and external barriers which hinder physical activity (PA) participation among physically inactive university employees in Tripoli. Their findings suggest that the primary barrier for poor PA participation was lack of time. Overall, it was interesting to look at some of the barriers addressed which limit physical activity participation and address different aspects of concern. The authors need to address these minor comments to improve readability.
Minor Comments
- Line 117-118: How was the screening questionnaire developed? Were there only a few questions in this questionnaire. From reading this sentence it look like there was only one question asked about whether or not they practice physical activity.
- Were the questionnaires validated?
- There is a limitation of gender distribution and there are more females than males in this population. The authors did mention in the introduction about gender differences so this distribution could bias the results.
- The discussion is currently very long. Please reduce the length and keep it focused on the main findings and make comparisons with previous studies.
- Please list some strengths and limitations. In the limitations, please address the gender distribution.
- Please address generalizability.
Author Response
Reviewer 1
Thank you for your valuable feedback and insightful comments on our manuscript. We appreciate your time and effort in reviewing our work. Below, we address each of your comments to improve the clarity and quality of our manuscript.
Comment 1: Line 117-118: How was the screening questionnaire developed? Were there only a few questions in this questionnaire. From reading this sentence it looks like there was only one question asked about whether or not they practice physical activity.
Response 1: Thank you for your insightful question regarding the development of the screening questionnaire. We would like to clarify that the questionnaire included one specific question asking participants whether they practice physical activity. This question serves as a screening measure to identify individuals who are physically inactive.
Comment 2: Were the questionnaires validated?
Response 2: Thank you for pointing this out. We would like to emphasize that the questionnaire utilized in our study is a well-established instrument commonly used in the literature to examine barriers to physical activity participation. It has been developed and validated in previous studies as mentioned in the manuscript. Additionally, we focused on measuring its test-retest reliability in our sample. This helps us ensure that the scores remain stable over time while drawing on the validation established by others. Therefore, this clarification is clearly articulated in the revised manuscript in the materials & methods section page 05 paragraph 02, to enhance understanding.
Comment 3: Were aspects associated with health considered for inclusion or exclusion in the study?
Response 3: Thank you for your thoughtful question. We would like to inform that this study didn't take participants' health status into consideration. We wanted to concentrate on demographic and occupational factors to better understand how they relate to the barriers to physical activity participation. While we recognize that health status can significantly affect outcomes, our goal was to focus on the other factors in this context. Therefore, we noted this point in the limitations of our study, suggesting that future research could benefit from looking into health-related aspects for a more complete understanding as shown in the revised manuscript, page 13 paragraph 03.
Comment 4: There is a limitation of gender distribution and there are more females than males in this population. The authors did mention in the introduction about gender differences so this distribution could bias the results
Response 4: We acknowledge the limitation regarding gender distribution in our sample. We revised the manuscript to explicitly discuss how the preponderance of female participants may influence the results and interpretations. Therefore, this was mentioned in the revised manuscript in both the discussion section (page 11 paragraph 01) and the limitations (page 13 paragraph 03).
Comment 5: The discussion is currently very long. Please reduce the length and keep it focused on the main findings and make comparisons with previous studies.
Response 5: We appreciate your feedback on the length of the discussion. We revised this section to ensure it is more concise and focused on the main findings, making clear comparisons with previous studies. Also, we removed unnecessary details to enhance readability as shown in the revised manuscript.
Comment 6: Please list some strengths and limitations. In the limitations, please address the gender distribution.
Response 6: We included a section summarizing both the strengths and limitations of our study. This encompasses the methodological rigor and the insights gained, as well as addressing the gender distribution issue as a limitation. page 13.
Comment 7: Please address generalizability.
Response 7: We added a paragraph in the discussion part on the generalizability of our findings, noting the context-specific nature of the study and the implications for broader populations. We emphasized that while the results may be applicable to similar university settings, caution should be exercised when generalizing to different contexts.
Thank you once again for your constructive comments. We believe that these revisions will significantly enhance the quality of our manuscript. We look forward to your further feedback.
Sincerely,
Dr. Sami Elmahgoub
corresponding author
Reviewer 2 Report
Comments and Suggestions for Authors
The study aims to characterize the barriers to physical activity in university employees. And to study the differences according to various variables such as age group and gender. Here are some suggestions.
Introduction:
Suggestion: Reinforce the relevance and innovation of the study.
Material and Methods:
Suggestion: Include the standard deviation of age in line 120.
Were aspects associated with health considered for inclusion or exclusion in the study?
Suggestion: Clarify who was included in the study. Only individuals who do not engage in organised physical activity. Clarify this in the article.
Suggestion: No ethical issues are reported. It is suggested that this information be integrated into this section.
Who applied the questionnaires and what training did they have?
Regarding the instrument:
What was the procedure for validating it for Libya? In which language was it administered? What is its reliability?
In ANOVA, how was the comparison between the three age groups performed? Was any multiple comparison test used? It would be interesting to conduct a MANCOVA.
Please indicate whether each dimension of the questionnaire is based on the average or sum of the questions.
Remove lines 202–207 after 'significance'.
As nutritional status is presented in Table 1, the methodology indicates how it was determined (reference values).
Results:
Suggestion: Include the n in the table in addition to the percentage.
In the tables, include the n.
Suggestion: In the description of the ANOVA results, also integrate statistics and effect.
In the results, also explore the value of the standard deviation and how it is interpreted.
Discussion:
Explore the limitations inherent to the study. Present possible lines of investigation. The discussion can be more fluid and concise.
References:
Check
Author Response
Reviewer 2
Thank you for your valuable feedback and insightful comments on our manuscript. We appreciate your time and effort in reviewing our work. Below, we address each of your comments to improve the clarity and quality of our manuscript.
Comment 1: Suggestion: Reinforce the relevance and innovation of the study.
Response 1: Thank you for pointing this out. We enhanced the introduction by emphasizing the significance of understanding barriers to physical activity among university employees, particularly in the context of rising sedentary lifestyles. We also highlighted the innovation of our research in addressing a gap in literature specific to Libya. All shown in page 02 paragraph 05 & page 03 paragraph 04.
Comment 2: Suggestion: Include the standard deviation of age in line 120.
Response 2: We included the standard deviation of age in the revised manuscript, to provide a clearer view of the age distribution in our sample. page 03 paragraph 05.
Comment 3: Were aspects associated with health considered for inclusion or exclusion in the study?
Response 3: Thank you for your thoughtful question. We would like to inform that this study didn't take participants' health status into consideration. We wanted to concentrate on demographic and occupational factors to better understand how they relate to the barriers to physical activity participation. While we recognize that health status can significantly affect outcomes, our goal was to focus on the other factors in this context. Therefore, we noted this point in the limitations of our study, suggesting that future research could benefit from looking into health-related aspects for a more complete understanding as shown in the revised manuscript, page 13 paragraph 04.
Comment 4: Clarify who was included in the study. Only individuals who do not engage in organized physical activity. Clarify this in the article.
Response 4: Thank you for your valuable feedback. We appreciate your request for clarification. In the manuscript, we have specified that participants were individuals who do not engage in physical activity in general, not just those who do not participate in organized physical activity programs. We ensured that this distinction is made clearer in the article to avoid any confusion.
Comment 5: No ethical issues are reported. It is suggested that this information be integrated into this section.
Response 5: Thank you for your insightful suggestion. We have included an ethical issues statement in the revised manuscript, detailing the ethical considerations relevant to our study. We appreciate your emphasis on this aspect and will ensure it is clearly integrated into the appropriate section for better visibility, page 04 paragraph 01.
Comment 6: Who applied the questionnaires and what training did they have?
Response 6: Thank you for your question! The questionnaires were administered by trained research assistants who underwent specific training before the study began. We’ve included this information in the revised manuscript to clarify how the administration process was handled, page 04 paragraph 03.
Comment 7: What was the procedure for validating it for Libya? In which language was it administered? What is its reliability?
Response 7: Thank you for pointing this out. We would like to emphasize that the questionnaire utilized in our study is a well-established instrument commonly used in the literature to examine barriers to physical activity participation. It has been developed and validated in previous studies as mentioned in the manuscript. The questionnaire was administered in Arabic language. Additionally, we focused on measuring its test-retest reliability in our sample. This helps us ensure that the scores remain stable over time while drawing on the validation established by others. Therefore, this clarification is clearly articulated in the revised manuscript in the materials & methods section page 05 paragraph 02, to enhance understanding.
Comment 8: In ANOVA, how was the comparison between the three age groups performed? Was any multiple comparison test used? It would be interesting to conduct a MANCOVA.
Response 8: Thank you for this valuable suggestion. In response, we have clarified the statistical analysis section to specify that post-hoc comparisons for ANOVA were conducted using Tukey’s HSD to control for multiple comparisons. Moreover, we performed a series of MANCOVAs to assess the multivariate effects of age, sex, occupation, and marital status on key dependent variables (energy, motivation, total internal and total external barriers). The MANCOVA findings, including Pillai’s Trace, F-values, degrees of freedom, and significance levels, have been integrated into the Results section. Updated tables summarizing these multivariate results have also been added to the manuscript.
Comment 9: Please indicate whether each dimension of the questionnaire is based on the average or sum of the questions.
Response 9: Thank you for your question. Each dimension of the questionnaire was calculated by taking the average (mean) of the responses to the individual questions. We have also included this information in the methods section of the manuscript to provide clarity, page 05 paragraph 03. We really appreciate your attention to this detail.
Comment 10: Remove lines 202–207 after 'significance'.
Response 10: Thank you for your suggestion. We have removed lines 202–207 after "significance" as requested.
Comment 11: As nutritional status is presented in Table 1, the methodology indicates how it was determined (reference values).
Response 11: Thank you for your valuable feedback. We assessed nutritional status using Body Mass Index (BMI). Since BMI is a familiar and widely recognized measure, we aimed to keep our explanation concise, respecting our readers' existing knowledge.
Comment 12: Include the n in the table in addition to the percentage. In the tables, include the n.
Response 12: Thank you for your feedback. We have included the sample size (n) in the tables, as suggested.
Comment 13: In the description of the ANOVA results, also integrate statistics and effect.
In the results, also explore the value of the standard deviation and how it is interpreted.
Response 13: Thank you for your suggestion. While the original analyses used ANOVA, we have now conducted and reported more comprehensive multivariate analyses (MANCOVAs) to account for the effects of age, sex, occupation, and marital status on multiple dependent variables simultaneously. These results include Pillai’s Trace, F-values, degrees of freedom, p-values, and have been summarized in newly added tables. We have also included interpretation of standard deviations in the Results section, especially for key variables such as time barriers, energy, and motivation, to reflect the variability in participants' responses and enhance the contextual understanding of the findings.
Comment 14: Explore the limitations inherent to the study. Present possible lines of investigation. The discussion can be more fluid and concise.
Response 14: Thank you so much for your helpful suggestions! In the revised manuscript, we’ve taken a closer look at the study's limitations and highlighted areas for improvement. We also included a new section called "Strengths and Limitations," along with possible lines of investigation for future research. Plus, we’ve worked to make the discussion more fluid and concise for easier reading.
Comment 15: References: Check
Response 15: Thank you for your attention to the references. We have checked them thoroughly, and one reference has been adjusted for accuracy.
Thank you once again for your constructive comments. We believe that these revisions will significantly enhance the quality of our manuscript. We look forward to your further feedback.
Sincerely,
Dr. Sami Elmahgoub
Corresponding author
Round 2
Reviewer 2 Report
Comments and Suggestions for Authors
The article has clearly improved. The suggestions presented have been considered. It is recommended that the formatting be checked.